# A Hybrid Quantum Image-Matching Algorithm

**DOI:** 10.3390/e24121816

**Published:** 2022-12-13

**Authors:** Guoqiang Shu, Zheng Shan, Shiqin Di, Xiaodong Ding, Congcong Feng

**Affiliations:** State Key Laboratory of Mathematical Engineering and Advanced Computing, Zhengzhou 450000, China

**Keywords:** image matching, amplitude encoding, quantum computing, amplitude estimation, quadratic acceleration

## Abstract

Image matching is an important research topic in computer vision and image processing. However, existing quantum algorithms mainly focus on accurate matching between template pixels, and are not robust to changes in image location and scale. In addition, the similarity calculation of the matching process is a fundamentally important issue. Therefore, this paper proposes a hybrid quantum algorithm, which uses the robustness of SIFT (scale-invariant feature transform) to extract image features, and combines the advantages of quantum exponential storage and parallel computing to represent data and calculate feature similarity. Finally, the quantum amplitude estimation is used to extract the measurement results and realize the quadratic acceleration of calculation. The experimental results show that the matching effect of this algorithm is better than the existing classical architecture. Our hybrid algorithm broadens the application scope and field of quantum computing in image processing.

## 1. Introduction

Image matching is one of the key technologies of digital photogrammetry in machine learning fields, such as information processing, pattern recognition, computer vision, etc. [1,2,3,4]. Image matching is known as the comparison of two images, and it can be roughly divided into three categories: the first category is based on image grayscale [5], the second category is related to image features [6,7], and the third category is in view of the understanding and interpretation of images. However, image matching faces some obstacles, such as object rotation, scaling and translation, illumination and noise, etc. Dong et al. proposed the DSP-SIFT (domain-size pooling scale-invariant feature transform) method that improves descriptor matching performance by pooling feature block size and gradient direction [8,9]. The local features and spatial layout are integrated and encoded into the edges of a graph whose nodes represent potential correspondences for discovering all common visual patterns. All strongly connected subgraphs correspond to large local maxima of a quadratic function on simplex using a replicator equation and through a systematic way of initialization. This was demonstrated to be robust to outliers, and able to discover all common visual patterns [10]. The method ICF (identifying point correspondences by correspondence function) was proposed for rejecting mismatches from given putative point correspondences. This introduces the correspondence function for capturing the relationships between corresponding points and an algorithm IECF (iteratively estimate correspondence function) based on the diagnostic technique and SVM. The method rejects the mismatches by checking whether they are consistent with the estimated correspondence functions, and demonstrated excellent performance [11]. The locality preserving matching (LPM) was proposed to seek reliable correspondences between two feature sets and the principle of which is to maintain the local neighborhood structures of those potential true matches. It achieves better or favorably competitive performance in accuracy while intensively cutting the time cost by more than two orders of magnitude [12]. The method (GMS) grid-based motion statistics was proposed to separate true correspondences from false ones at high speed, which incorporates the smoothness constraint into a statistic framework for separation and uses a grid-based implementation for fast calculation. GMS is robust to various challenging image changes [13]. There may leave several open questions about which method would be a suitable choice for specific applications with respect to different scenarios and task requirements and how to design better image-matching methods with superior performance inaccuracy, robustness and efficiency. A work analyzed them from the perspective of theory, experiment, etc. [14]. A method motion-consistency driven matching (MCDM) for mismatch removal was proposed, which formulates the matching problem into a probabilistic graphical model to infer the state of each node. The final inference is cast into an integer quadratic programming problem, and the solution is obtained by using an efficient optimization technique based on the FrankWolfe algorithm. The strong generalization ability as well as high accuracy, which outperforms state-of-the-art methods, was demonstrated in the work [15]. The multi-scale attention-based network (called MSA-Net) feature-matching method was proposed to solve the limited effectiveness and robustness when applied to different scenarios. By introducing a multi-scale attention block to enhance the robustness to outliers, it is able to effectively infer the probability of correspondences being inliers with fewer parameters, achieving a 11.7% improvement at error threshold 5 compared to the state-of-the-art method on the YFCC100M dataset [16].

Furthermore, the similarity calculation methods of features are diverse. For example, the correlation function, covariance function, structural similarity index method, etc. [17,18]. Moreover, high accuracy is necessary for image matching, and thus the amount of calculation is huge, so it is difficult to meet the requirements of real-time operation, and the matching rate will increase in areas with poor information frequency. Fortunately, advances in quantum information science and machine learning have led to the natural emergence of quantum machine learning, a field that bridges the two, aiming to revolutionize information technology [19,20,21]. The core of its interest lies in either taking advantage of quantum mechanics to achieve machine learning that surpasses the classical pendant in terms of computational complexity or to entirely be able to apply such techniques on quantum data. An important application of machine learning is pattern recognition in big data analysis [22].

There have been several references that discuss the quantum image matching. A pattern-matching method based on the Grover algorithm is proposed, and approximate amplitude coding and inversion test operation are used. Experiments show that the algorithm is effective; however, the dimension requirements for querying images are low [23]. Daniel Curtis et al. considered a problem of locating a template as a sub image of a larger image [24]. Algorithms that search for a pattern within a larger dataset was proposed; the complexity of the theory is $O(\sqrt{n})$ [25]. They used Fourier transform to calculate the correlation, and the computational complexity was found to be less than the classical correlation algorithm. Yang et al. gave a quantum gray-scale image matching scheme [26]. By mapping the template image with the quantum reference image directly, the scheme is given to calculate the difference of all pixels in two images. Furthermore, Luo et al. proposed a quantum fuzzy matching algorithm based on the pixel-mapping difference, which shows that quantum is exponential parallel acceleration [27]. Jiang et al. investigated the quantum image matching in depth [28,29]. They proposed a quantum matching scheme based on Grover algorithm to find out a small image in a big image. Unfortunately, the paper only matched one pixel and without more than one pixel, the algorithm will cause an error.Therefore, an improved version was presented which takes full advantage of the whole matched area to locate a small image in a big image soon after that [30]. Zhou et al. proposed a quantum image similarity analysis method based on the combination of quantum image pixel representation and amplitude amplification, which demonstrates the strong parallel ability and secondary acceleration advantages of quantum compared with the classical violent element search method [31]. In addition, Ashley proposed a quantum pattern matching algorithm based on pixel, which is super-polynomially faster than the classical algorithm [32].

Previous research work is mainly based on the image pixel matching method. This requires high consistency between template images and reference images, and greatly limits the expansion of matching applications. Therefore, this paper proposes a hybrid quantum image-matching algorithm based on feature similarity. The classical SIFT algorithm is used to extract the feature attributes of the image and effectively solve the problem of matching error caused by image deformation and other factors. At the same time, we combine the exponential storage and parallel computing capabilities of quantum to calculate the similarity of features, which improves the efficiency of computing and reduces the consumption of resources greatly. Finally, the quantum amplitude estimation algorithm is applied to bring quadratic acceleration to the measurement process of the results.

The article is organized as follows. Section 2 introduces the hybrid quantum algorithm, including the classical feature extraction method, quantum image representation, quantum similarity calculation and amplitude estimation algorithm. In Section 3, two datasets are selected to test the effectiveness of the proposed algorithm, and we analyze the experimental results compared with the classical algorithm. Finally, the conclusions are drawn in Section 4.

## 2. Materials and Methods

### 2.1. Feature Extraction and Matching

The SIFT detector extracts a number of meaningful descriptive features [33,34,35]. These features are invariant to scaling, rotation, and illumination. Such points are generally presented in high-contrast areas, possibly on object edges. One significant quality of these features is that the relative positions between them should not change from one image to another.

The main stage is scale–space extrema extraction are as follows. The first stage is scale–space extrema extraction. In this stage, the interest points, which are scale and rotation invariant are searched, and are obtained with the difference of Gaussian (DoG) function. Then, the key point localization and filtering are performed. In this stage, the location and scale for resultant interest points are found, and the key points are selected, which are robust to image distortion. The next stage is the orientation assignment in which one or more orientation is assigned to each key point location based on the local image-gradient directions. Finally, the feature description is performed. The local image gradients are measured at the selected scale in the neighborhood of a key point, and the 128D feature descriptor is obtained (Figure 1).

Quantum is characterized by parallelism, as is classical computing. For the same number of bits *n*, the data representation and processing capability of the quantum $O(2^{n})$ is beyond the classical computing capability O(n). Hence, we propose to analyze the similarity of features with quantum, as shown in Figure 2. Here is a diagram of the hybrid algorithm, where each colored small grid represents a feature point, and the color category stands for the category of the feature point.

After the descriptors are obtained, the similarity between them needs to be calculated. There are many classical methods of similarity calculation for feature points matching [36,37]. For example, the correlation function is a method to search for a window *Y* whose correlation function value with window *X* reaches a maximum value within a certain search range D, and it can be equivalent to the inner product operator and cosine distance:(1)$$R\left(X,Y\right)=\sum_{i=1}^{m}\sum_{j=1}^{n}x_{ij}y_{ij},\;Y_{\max}=\max_{Y\in D}R\left(X,Y\right)$$

### 2.2. The Quantum Encoding of Arbitrary Initial State

Many results in quantum information theory require the generation of specific quantum states, or the implementation of encoding a given vector onto qubits for computing, such as the quantum swap-test similarity calculation. However, many states cannot be efficiently realized. There are many methods focusing on the generation of quantum states, for example, Huelga et al. implemented an improved frequency standard experiment that requires the preparation of specific symmetric states on *n* qubits, where *n* is a parameter (number of ions) [38]. Kaye et al. proposed an algorithm, efficiently preparing the required symmetric state [39]. For any normalized vector $X_{i}$, $X_{i}=\left(x_{0i},x_{1i},\cdots ,x_{n-1i}\right)$. We suppose that $n=2^{k}$; otherwise, the vector can be filled with 0. Long et al. proposed an algorithm for preparing arbitrary superposition quantum states with time complexity $O(n{\left(\log n\right)}^{2})$, space complexity O((logn)) and no additional auxiliary bits required [40]. The general method of encoding quantum states is shown in Figure 3.

Assuming that the probability corresponding to each basis is $p_{i}$, $p_{i}=\frac{{\left|x_{0i}\right|}^{2}}{{∥X_{i}∥}^{2}}$ the number of required qubit is *k*. When the first qubit is 0, the probability is $p_{0}=\sum_{i=0}^{2^{k-1}-1}p_{i}=\cos\theta_{00}^{2}$, which is 1, $p_{1}=\sum_{i=2^{k-1}}^{2^{k}-1}p_{i}=\sin\theta_{00}^{2}$. It can just be represented by the quantum gate
(2)$$Ry\left(2\theta_{00}\right)=e^{-i\theta_{00}\sigma_{y}}=\left[\begin{array}{cc} \cos\theta_{00} & -\sin\theta_{00}\\ \sin\theta_{00} & \cos\theta_{00} \end{array}\right]$$

When the second qubit is 0, the probability is
(3)$$\begin{array}{c} p_{00}=\sum_{i=0}^{2^{k-2}-1}p_{i}=\cos\theta_{00}^{2}\cos\theta_{10}^{2},\\ p_{01}=\sum_{i=2^{k-2}}^{2^{k-1}-1}p_{i}=\cos\theta_{00}^{2}\sin\theta_{10}^{2} \end{array}$$

When the second qubit is 1, the probability is
(4)$$\begin{array}{c} p_{10}=\sum_{i=2^{k-1}}^{2^{k-1}+2^{k-2}-1}p_{i}=\sin\theta_{00}^{2}\cos\theta_{11}^{2},\\ p_{11}=\sum_{i=2^{k-1}+2^{k-2}}^{2^{k}-1}p_{i}=\sin\theta_{00}^{2}\sin\theta_{11}^{2} \end{array}$$

Repeating the previous steps according to the rule of dichotomy until all k-bit values are traversed, the circuit in Figure 3 can be obtained. In addition, for a vector set *S* of *m* elements, $S=\{X_{0},X_{1},\cdots ,X_{m-1}\}$, where $X_{i}$ is a n-dimensional vector defined as before. Without loss of generality, we can set *m* is an integer power of 2, $m=2^{l}$, thus an equal probability quantum superposition state |ϕ〉, including labels about these vectors $|X_{0}\rangle$, $|X_{1}\rangle$⋯, $|X_{m-1}\rangle$, can be constructed. The number of qubit is ⌈logn⌉+⌈logm⌉, and the circuit time complexity is $O(\left(mn\right){\left(\log_{2}\left(m\right)\log_{2}\left(n\right)\right)}^{2})$, where the ⌈logm⌉ qubits are used as the label for state |ϕ〉. The quantum circuit is shown in Figure 4. When the dimensions of the elements in *S* are inconsistent, the number of bits required is inconsistent. In this case, we take the number of bits required by the vector with the largest dimension as the number of bits for each $U_{i}$ circuit:(5)$$\begin{array}{c} |X_{i}\rangle =A_{1}{|0\rangle }^{⊗k}=\frac{1}{\sqrt{x_{0i}^{2}+x_{1i}^{2}+\cdots +x_{n-1i}^{2}}}\sum_{k=0}^{n-1}|x_{ki}||k\rangle \end{array}$$(6)$$\begin{array}{c} |\phi \rangle =A_{2}{|0\rangle }^{⊗(k+l)}=\frac{1}{\sqrt{m}}\sum_{i=0}^{m-1}\left|i\rangle \right|X_{i}\rangle \end{array}$$
where $A_{1}$ and $A_{2}$ represents the unitary operations in Figure 3 and Figure 4 respectively, and $U_{i}$ stands for the operation to prepare quantum state $|X_{i}\rangle$.

### 2.3. Similarity Calculation Based on Swap Test

For comparing the similarity of two states |ϕ〉 and |ψ〉, where |ψ〉 is a superposition state consists of *pn*-dimensional vectors, and the quantum swap-test circuit is available [41,42]:(7)$$\begin{array}{c} |\psi \rangle =\frac{1}{\sqrt[{}]{p}}\sum_{i=0}^{p-1}\left|i\rangle \right|Y_{i}\rangle ,\;|Y_{i}\rangle =\frac{1}{\sqrt{y_{0i}^{2}+y_{1i}^{2}+\cdots +y_{n-1i}^{2}}}\sum_{k=0}^{n-1}|y_{ki}||k\rangle \end{array}$$

The result can be obtained by measuring the first register and the labels of state |ϕ〉 and |ψ〉, as shown in Figure 5. The measurement probability is $P_{0}=\left(1+|\langle \phi |\psi \rangle {|}^{2}\right)/2$, and $P_{1}=1-P_{0}$. Where $P_{0}$ represents the probability when the measuring result is |0〉. Thus, the fidelity-based similarity of vectors ψ and ϕ can be obtained by $\langle \phi |\psi \rangle =\sqrt{\left(2P_{0}-1\right)}$.
(8)P0=14∥(I+SWAP)(Q|0〉|ψ〉|ϕ〉)∥2=1+Re(〈ψ⊗ϕ∣SWAP∣ψ⊗ϕ〉)2=1+∣〈ϕ∣ψ〉∣22
and $Q=\left(H⊗I^{⊗n}\right)\left(C-SWAP\right)\left(H⊗I^{⊗n}\right)$ represents the operator of the swap test.
(9)$$\begin{array}{cc} U_{swap}|0\rangle |\psi \rangle |\phi \rangle & =\frac{|0\rangle +|1\rangle }{2}|\psi ⊗\phi \rangle +\frac{|0\rangle -|1\rangle }{2}SWAP|\psi ⊗\phi \rangle \\ \; & =\frac{1}{\sqrt{mp}}\sum_{i}^{m-1}\sum_{j}^{p-1}(|0\rangle |ij\rangle \frac{|X_{i}⊗Y_{j}\rangle +SWAP|X_{i}⊗Y_{j}\rangle }{2}\\ \; & \;+|1\rangle |ij\rangle \frac{|X_{i}⊗Y_{j}\rangle -SWAP|X_{i}⊗Y_{j}\rangle }{2}) \end{array}$$

The measurement probability of quantum state $|0\rangle |ij\rangle (|Y_{j}⊗X_{i}\rangle +SWAP|Y_{j}⊗X_{i}\rangle )$ is
(10)P0ij=14mp∥(I+SWAP)(Q|0〉|Yj〉|Xi〉)∥2=1+Re(〈Xi⊗Yj∣SWAP∣Xi⊗Yj〉)2mp=1+∣〈Xi∣Yj〉∣22mp
when the first register is |0〉 and we can obtain the probability $P_{0}^{ij}$ by measuring the labels |i〉 and |j〉, i.e., similarity of $\langle X_{i}|Y_{j}\rangle =\sqrt{\left(2mp*P_{0}^{ij}-1\right)}$. For the swap operation, it can be decomposed as three CNOT gates, thus the swap test can be represented as the gate set of Hadamard and CNOT. A simplified equivalent circuit for estimating the expectation value of the swap-test operation is proposed in [43], and the circuit employs one CNOT gate followed by one Hadamard gate, with both qubits being measured, which diagonalizes the swap gate, when it is treated as a Hermitian observable with eigenvalues 1 or −1 (Figure 6). The output distribution that constructs an estimator for tested states |ϕ〉 and |ψ〉 is $|\langle \psi |\phi \rangle {|}^{2}$.

For example, when $|\phi \rangle =\alpha_{1}|0\rangle +\beta_{1}|1\rangle$, $|\psi \rangle =\alpha_{2}|0\rangle +\beta_{2}|1\rangle$, after a CNOT and Hadamard operation, the result is
(11)$$\frac{1}{\sqrt{2}}[\left(\alpha_{1}\alpha_{2}+\beta_{1}\beta_{2}\right)|00\rangle +\left(\alpha_{1}\alpha_{2}-\beta_{1}\beta_{2}\right)|10\rangle +\left(\alpha_{1}\beta_{2}+\beta_{1}\alpha_{2}\right)|01\rangle +\left(\alpha_{1}\beta_{2}-\beta_{1}\alpha_{2}\right)\left|11\rangle \right]$$

The probability of measuring the |11〉 state is
(12)$$P_{11}=\frac{|\alpha_{1}\beta_{2}-\beta_{1}\alpha_{2}{|}^{2}}{2}=\frac{\left(\alpha_{1}\beta_{2}-\beta_{1}\alpha_{2}\right){\left(\alpha_{1}\beta_{2}-\beta_{1}\alpha_{2}\right)}^{*}}{2}$$

The probability amplitudes in the input qubits obey $|\alpha_{1}{|}^{2}+|\beta_{1}{|}^{2}=1$ and $|\alpha_{2}{|}^{2}+|\beta_{2}{|}^{2}=1$.
(13)$$\begin{array}{cc} 1-P_{11} & =\frac{2-|\alpha_{1}{|}^{2}|\beta_{2}{|}^{2}-|\beta_{1}{|}^{2}|\alpha_{2}{|}^{2}+\alpha_{1}\beta_{1}^{*}\alpha_{2}^{*}\beta_{2}+\alpha_{1}^{*}\beta_{1}\alpha_{2}\beta_{2}^{*}}{2}\\ \; & =\frac{1+|\alpha_{1}{|}^{2}|\alpha_{2}{|}^{2}+|\beta_{1}{|}^{2}|\beta_{2}{|}^{2}+\alpha_{1}\beta_{1}^{*}\alpha_{2}^{*}\beta_{2}+\alpha_{1}^{*}\beta_{1}\alpha_{2}\beta_{2}^{*}}{2} \end{array}$$

Because of the following relationship
(14)$$\begin{array}{cc} |\langle \psi |\phi \rangle {|}^{2} & =\left(\alpha_{1}\alpha_{2}^{*}+\beta_{1}\beta_{2}^{*}\right)\left(\alpha_{1}^{*}\alpha_{2}+\beta_{1}^{*}\beta_{2}\right)\\ & =|\alpha_{1}{|}^{2}|\alpha_{2}{|}^{2}+|\beta_{1}{|}^{2}|\beta_{2}{|}^{2}+\alpha_{1}\beta_{1}^{*}\alpha_{2}^{*}\beta_{2}+\alpha_{1}^{*}\beta_{1}\alpha_{2}\beta_{2}^{*} \end{array}$$
the probability of the swap test is $P=\frac{1+|\langle \psi |\phi \rangle {|}^{2}}{2}$ corresponds to that in Equation (Equation 13).

### 2.4. Extraction of Measurement Results

The results of quantum computing are in the amplitude of the superposition state. To obtain these results, the most direct method is to perform a large number of measurements, but it has no acceleration advantage, so the similarity calculation based on amplitude amplification and amplitude estimation can bring a quadratic acceleration advantage. Let |Ψ〉=A|0〉 denote the state obtained by applying A to the initial zero state. The amplification process is realized by repeatedly applying the following unitary operator [44,45] on the state |Ψ〉:(15)$$Q=Q\left(A,\chi \right)=-AS_{0}A^{-1}S_{\chi }$$

Here, the operator $S_{\chi }$ conditionally changes the sign of the amplitudes of the good states,
$$|x\rangle ⟼\left\{\begin{array}{cc} -|x\rangle & \;\mathrm{if}\;\chi (x)=1\\ |x\rangle & \;\mathrm{if}\;\chi (x)=0, \end{array}\right.$$
while the operator $S_{0}$ changes the sign of the amplitude if and only if the state is the zero state |0〉. The operator $U_{swap}$ is well-defined since we assume that A uses no measurements and, therefore, A has an inverse. The usefulness of operator $U_{swap}$ stems from its simple action on the subspace $H_{\Psi }$ spanned by the vectors $|\Psi_{1}\rangle$ and $|\Psi_{0}\rangle$. If we want to obtain the amplitude of $|0\rangle |ij\rangle (|X_{i}⊗Y_{j}\rangle +SWAP|X_{i}⊗Y_{j}\rangle )$, $\sqrt{P_{0}^{ij}}$, then the amplitude estimation is available, and we can write the result as
(16)$$\begin{array}{cc} |\gamma \rangle & =U_{swap}A_{2}A_{1}|0\rangle =\sqrt{P_{0}^{ij}}(|0\rangle |ij\rangle (|X_{i}⊗Y_{j}\rangle +SWAP|X_{i}⊗Y_{j}\rangle ))\\ \; & \;+\sqrt{1-P_{0}^{ij}}(\sum_{s=0,s\neq i}^{m-1}\sum_{t=0,t\neq j}^{p-1}|0\rangle |st\rangle (|X_{s}⊗Y_{t}\rangle +SWAP|X_{s}⊗Y_{t}\rangle )\\ \; & \;+\sum_{s=0}^{m-1}\sum_{t=0}^{p-1}|1\rangle |st\rangle (|X_{s}⊗Y_{t}\rangle -SWAP|X_{s}⊗Y_{t}\rangle )) \end{array}$$

If we remark that $\tilde{A}=U_{swap}A_{2}A_{1}$, $\tilde{Q}=-\tilde{A}S_{0}{\tilde{A}}^{-1}S_{\chi }$, $\sqrt{P_{0}^{ij}}=\sin\theta /2$, θ∈[0,π], and we can obtain
(17)$$\begin{array}{cc} |\gamma \rangle & =\tilde{A}|0\rangle |0\rangle |0\rangle =\sin\theta /2(|0\rangle |ij\rangle (|X_{i}⊗Y_{j}\rangle +SWAP|X_{i}⊗Y_{j}\rangle ))\\ \; & \;+\cos\theta /2(\sum_{s=0,s\neq i}^{m-1}\sum_{t=0,t\neq j}^{p-1}|0\rangle |st\rangle (|X_{s}⊗Y_{t}\rangle +SWAP|X_{s}⊗Y_{t}\rangle )\\ \; & \;+\sum_{s=0}^{m-1}\sum_{t=0}^{p-1}|1\rangle |st\rangle (|X_{s}⊗Y_{t}\rangle -SWAP|X_{s}⊗Y_{t}\rangle )) \end{array}$$

$S_{\chi }\left|\gamma \rangle =-\sin\theta /2\right|\gamma_{1}\rangle +\cos\theta /2|\gamma_{0}\rangle$, and after *k* interactions, the result is
(18)$${\tilde{Q}}^{k}|\gamma \rangle =\cos\left(\left(2k+1\right)\theta \right)|\gamma_{0}\rangle +\sin\left(\left(2k+1\right)\theta \right)|\gamma_{1}\rangle$$

By amplitude estimation (Figure 7), we can obtain an estimation value of $\tilde{\theta }$. Thus $\sqrt{P_{0}^{ij}}=\sin\tilde{\theta }/2$, where *k* is an integer power of 2 with respect to *t*. Thus, we can obtain the similarity calculation of each feature vector of the target image and the feature vector of matching images, and it can bring a quantum quadratic acceleration advantage compared with direct measurement.

## 3. Results and Discussion

For exhibiting the effectiveness of the hybrid quantum image matching algorithm, we implemented the experiments based on an open NWPU-RESISC45 dataset, a benchmark for remote sensing image scene recognition created by Northwestern Polytechnical University, and another open handwritten character dataset, Mnist. We used the SIFT feature points of these datasets in experiments, and applied classical cosine similarity and the quantum swap test for calculation. An example of image matching is shown in Figure 8. Figure 8a represents the template image; Figure 8b, the references to be matched; and Figure 8c, the result of the feature matching.

Experiments are implemented with IBM qasm−simulator, which supports 32-qubit and limited circuit depth. When the number of feature points is small, we apply the circuit in Figure 4; otherwise, the circuit in Figure 3 can be used to prepare the state one by one. The similarity is calculated according to the ratio between the nearest neighbor distance $d_{1}\left(R_{i},S_{i}\right)$, and the next nearest neighbor distance $d_{2}\left(R_{i},S_{j}\right)$ is less than a certain threshold. The experiments use the cosine distance, and we obtain quantum and classical results under different thresholds, respectively:(19)$$\frac{d_{1}\left(R_{i},S_{i}\right)}{d_{2}\left(R_{i},S_{j}\right)}<threshold$$

There are two evaluation indicators of the matching system effect, recall and precision. The recall represents the proportion of matched feature points among all feature points, which stands for the recall rate of the system. The precision stands for the proportion of correctly matched feature points among all the matched feature points, which represents the matching accuracy of the matching system:(20)$$\begin{array}{c} \mathrm{Recall}=\frac{|\mathrm{correct}\mathrm{matches}|}{|\mathrm{correspondences}|} \end{array}$$
(21)$$\begin{array}{c} \mathrm{Precision}=\frac{|\mathrm{correct}\mathrm{matches}|}{|\mathrm{correct}\mathrm{matches}|+|\mathrm{false}\mathrm{matches}|} \end{array}$$
where ∣·∣ represents the number of elements in the set, and *correspondences* stands for the set of all feature points that need to be matched in template images.

Experimental results and analysis: Figure 9 is the result of NWPU-RESISC45 dataset. From Figure 9a–c, the horizontal axis represents the threshold, and the vertical axis corresponds to the values of recall, precision and F1, respectively. In the experiments, the quantum results are obtained by random sampling of the measured results. Due to the influence of the sampling accuracy, the number of misidentified points in the experiments will be more than those of the classicals. Under the same threshold, the recall of quantum swap-test exhibits better performance than that of the classicals. As the threshold increases, the matching accuracy of similarity calculation based on classical cosine similarity and quantum swap-test also increases. That is because the upper bound of the ratio between the nearest neighbor and the next nearest neighbor becomes larger as the threshold increases, which leads to an increase in the number of mismatched points, and thus the recall rate (Figure 9a).

Obviously, the greater the number of misidentified points, the lower the accuracy. Thus, the quantum precision is slightly lower than classical precision, as shown in Figure 9b. Precision and recall are inherently contradictory; therefore, a comprehensive evaluation index F-score is required, which is the average of the weighted sum of precision and recall. When α=1, it is called the F1-score.
(22)$$\begin{array}{c} F=\frac{(\alpha^{2}+1)P*R}{\alpha^{2}\left(P+R\right)} \end{array}$$

The larger the value of F, the better the matching effect. The trend line in Figure 9c shows that the hybrid algorithm based on the swap test is better than the classical cosine similarity calculation method.

Similar to Figure 9, Figure 10a–c in Figure 10 stands for recall, precision, and F1, in turn. The results show that the same conclusion can be obtained on the Mnist dataset, which proves that the hybrid algorithm is better than the classical algorithm again.

## 4. Conclusions

By conducting a series of experiments based on classical and our hybrid methods, some useful conclusions can be achieved:(1)The validity and accuracy of the hybrid method based SIFT are proved through experimental verification, and it was also demonstrated that the hybrid algorithm performs better than the classicals.(2)By the amplitude encoding, the quantum resources consumption is logarithmically less than that of classical resources, making the space complexity exponentially reduced. The computing resources can be reduced from $O(mpn^{2})$ to O(2log(n)+logp+logm).(3)Making use of quantum entanglement and superposition properties, the time complexity of the similarity calculation between two images with *s* and *t* feature descriptors is reduced from O(st) to O(1). Additionally, the value of st is usually very large.(4)The acquisition of quantum computing results requires measurement, and direct measurement does not show the quantum advantage. Thus, it is possible to obtain the similarity by estimating value of θ, $\sin^{2}\tilde{\theta }/2=P_{0}^{ij}$, which brings quadratic acceleration advantage $O(\sqrt{mp})$.

This paper theoretically proves the quadratic acceleration advantage of the quantum hybrid algorithm and logarithmic resource consumption advantage over classicals. At the same time, it is also experimentally proved that the comprehensive performance of the hybrid algorithm performs better than that of the classical algorithm, showing that the quantum hybrid algorithm is used in machine learning, especially the great potential in the field of image matching. With the development of quantum computers with large number of bits, high bit connectivity, and long decoherence time, the limitations of massive data calculations in the research fields, such as images processing, are eliminated by quantum computing.

## Figures and Tables

**Figure 1 entropy-24-01816-f001:**
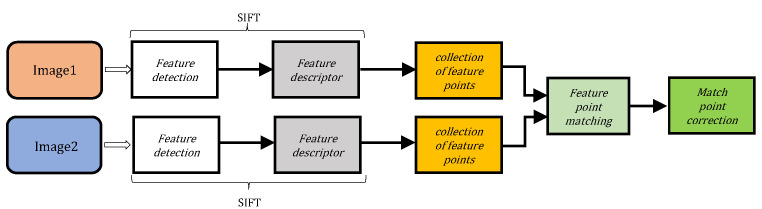
Diagram of SIFT feature extraction.

**Figure 2 entropy-24-01816-f002:**
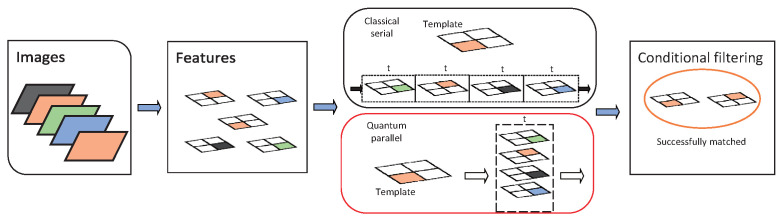
Flow chart of matching algorithm. The part contained in the red box is quantum substitution for classical calculations in black; it is also the structure of the hybrid matching algorithm.

**Figure 3 entropy-24-01816-f003:**
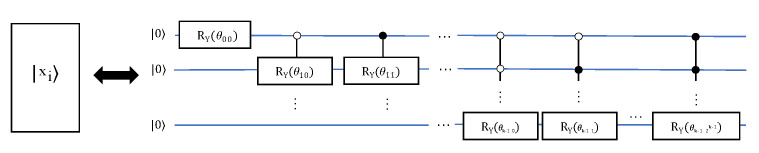
Diagram of encoding a vector to the required quantum state.

**Figure 4 entropy-24-01816-f004:**
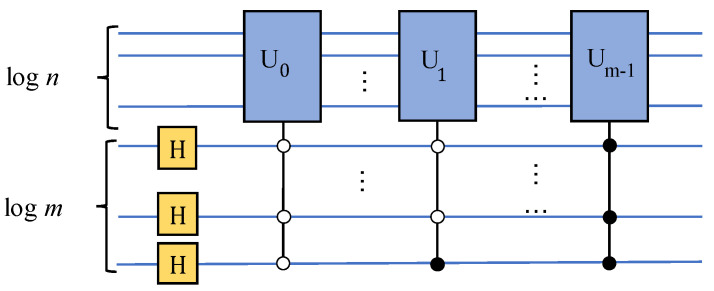
Diagram of constructing a equal probability quantum superposition state.

**Figure 5 entropy-24-01816-f005:**
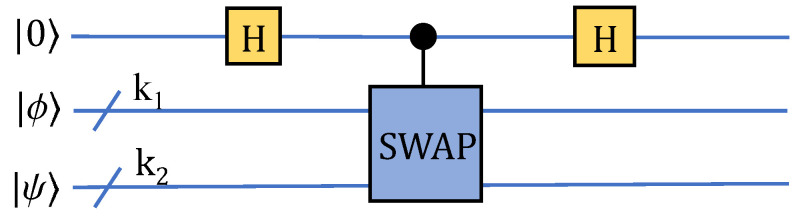
Diagram of similarity calculation based on fidelity.

**Figure 6 entropy-24-01816-f006:**
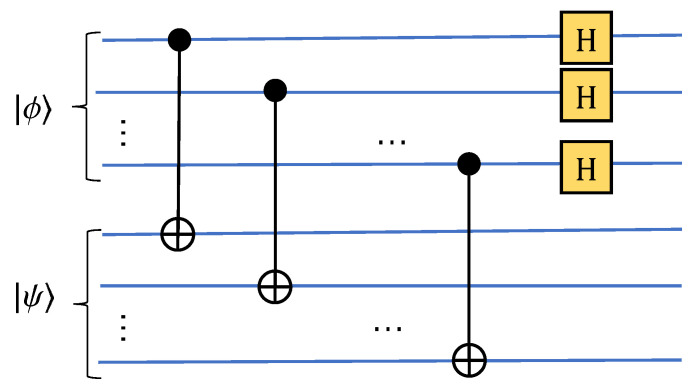
Diagram of simplified swap-test circuit.

**Figure 7 entropy-24-01816-f007:**
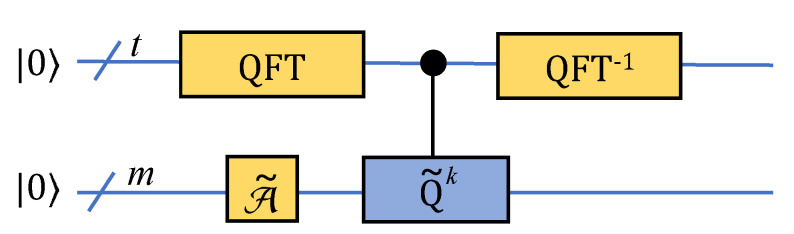
Diagram of amplitude estimation.

**Figure 8 entropy-24-01816-f008:**
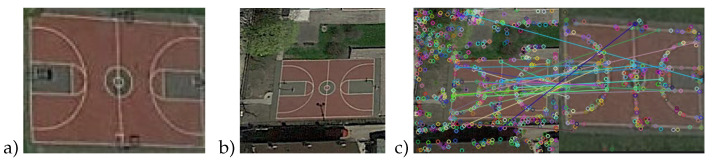
Example of matching. (**a**) This is template image; (**b**) this is the reference to be matched; (**c**) this is result of feature matching.

**Figure 9 entropy-24-01816-f009:**
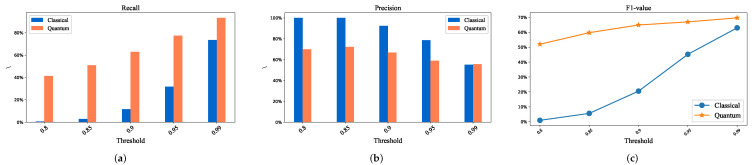
The comparative experimental results of classical and hybrid quantum method based on NWPU-RESISC45 dataset. (**a**) Relationship between threshold and Recall. (**b**) Relationship between threshold and Precision. (**c**) Relationship between threshold and F1-value.

**Figure 10 entropy-24-01816-f010:**
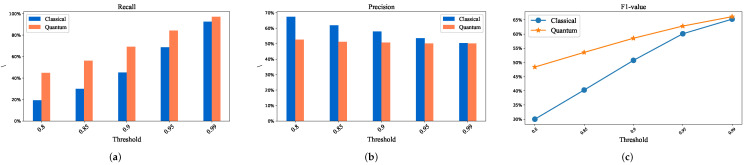
The comparative experimental results of classical and hybrid quantum method based on Mnist dataset. (**a**) Relationship between threshold and Recall. (**b**) Relationship between threshold and Precision. (**c**) Relationship between threshold and F1-value.

## Data Availability

All relevant data supporting the main conclusions and figures of the document are available on request. Please refer to Zheng Shan at zzzhengming@163.com.

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
