# Peer review of "A Hybrid Quantum Image-Matching Algorithm"

_entropy, 2022, doi:10.3390/e24121816_

Round 1
Reviewer 1 Report
This paper proposes a hybrid quantum image matching method. The only contribution is to introduce quantum to image matching.
There are many serious problem:
1. The related work is almostly ignored.
2. The abstract and introduction are required to be re-organized. That is, the motivation and contributions are not clear.
3. The experiment is also not convincing. The datasets are very small. There are not one competting methods!
Reviewer 2 Report
as attached file

Reviewer 3 Report
The goal of this research is image matching. A classical and quantum hybrid algorithm is proposed, which is based on classical feature matching and quantum similarity measurement method. The algorithm shows quadratic acceleration ability and the effectiveness is verified in the experiment. Having said that, I still have some comments and suggestions.
Point 1: The manuscript proposes a hybrid quantum algorithm and points out in the abstract that the algorithm has lower complexity. But it does not mention how to reduce the complexity.
Point 2:Figure 3 shows that the vector set is represented as a labeled superposition states. Whether the method is still feasible when the dimensions of vectors are different needs further analysis and discussions.
Point 3:In the part of experimental results, the reasons for the differences of results under different methods should be analyzed in more detail.
Point 4: The number of keywords can be further increased to reflect the main methods and characteristics of the research.
Point 5: In the representation of the next line of Eq. 8, the superscript of I should include n.
Point 6: The case of some words should be consistent, for example: ‘Swap-test’.
Round 2
Reviewer 1 Report
I don't think the authors have carefully considered my comments.
Firstly, the related work is also ignored. From the references, there is only one paper from 2021, and zero from 2022. Most of them are very old. Image matching is an important task for computer vision, and there are many good papers.
1. Feature Matching via Motion-Consistency Driven Probabilistic Graphical Model, International Journal of Computer Vision (IJCV), 2022, 130 (9), 2249-2264
2. Locality preserving matching. International Journal of Computer Vision, 127(5), 512–531
3. Image matching from handcrafted to deep features: A survey. International Journal of Computer Vision, 129(1), 23–79.
4.Gms: Grid-based motion statistics for fast, ultra-robust feature, International Journal of Computer Vision 128 (2020) 1580–1593.
5. Common visual pattern discovery via spatially coherent correspondences, in: IEEE Conference on Computer Vision and Pattern Recognition
6. Rejecting mismatches by correspondence function, International Journal of Computer Vision
7. MSA-Net: Establishing Reliable Correspondences by Multi-Scale Attention Network, IEEE Transactions on Image Processing (TIP), 2022
All of them are opensource. They are only a part of the papers about image matching.
Secondly, the experiments are also not convinced. The competing methods are also been ignored. Some public datasets, such as, VGG, DAISY, Retina and EVD, are not been mentioned.
Reviewer 3 Report
The manuscript is suitable for publication.
Author Response
Thank you for your sincere comments!
Round 3
Reviewer 1 Report
The authors have considered most of my comments. Still, most of related work are based on deep learning. So, if the authors can add more other references, it will be better. Here, we provide some suggestions (it is not neccessary):
Robust Feature Matching for Remote Sensing Image Registration via Guided Hyperplane Fitting, Guobao Xiao, Huan Luo, Kun Zeng, Leyi Wei and Jiayi Ma, IEEE Transactions on Geoscience and Remote Sensing, 2022, 60, 1-14
Segmentation by Continuous Latent Semantic Analysis for Multi-structure Model Fitting, Guobao Xiao, Hanzi Wang, Jiayi Ma and David Suter
International Journal of Computer Vision, 2021, 129, 2034-2056
Mining consistent correspondences using co-occurrence statistics,Guobao Xiao, Shiping Wang, Han Wang and Jiayi Ma,
Pattern Recognition, 2021, 119, 108062
Superpixel-guided two-view deterministic geometric model fitting, Guobao Xiao, Hanzi Wang, Yan Yan, David Suter
International Journal of Computer Vision, 2019, 127 (4), 323-339